# RHD6LA regulates root hair responses to both symbionts and commensals

Francesca Tedeschi [1,2] ✉, Johan Quilbé [1,2], Lavinia Ioana Fechete [1], Sofie Jin Vistisen Christiansen[1] & Stig Uggerhøj Andersen [1] ✉

While intracellular symbiosis with rhizobia relies on Nod factor signaling through the conserved common symbiosis signaling pathway (CSSP), it remains unclear how legumes simultaneously manage interactions with commensal soil microbes. Using single cell RNA-sequencing, we show that commensal soil bacteria induce a Nod factor-independent transcriptional response in specific root hairs. This response is similar to the rhizobium response in the CSSP-deficient *cyclops* mutant, which is unable to accommodate rhizobia in root hair infection threads. Both responses include the nodulation gene *NODULATION SIGNALING PATHWAY 2* (*NSP2*) and a transcription factor, which we name *ROOT HAIR DEFECTIVE 6 LIKE A* (*RHD6LA*). We show that *RHD6LA* is required for facilitating infection thread formation in response to rhizobia and for preventing exaggerated root hair responses to commensal soil bacteria. The overlap between commensal and symbiotic signaling highlights the complexity of legume-microbe interactions at the root hair interface and suggests additional mechanisms for microbial discrimination in rhizobium-responsive root hairs.

Plant roots harbor a diverse community of bacteria known as the root microbiota, which play an important role in shaping plant growth, health, and overall ecosystem function[1–4]. This association between plants and their root-associated bacteria has gained increasing attention due to its potential implications for agriculture and environmental sustainability[5–7]. However, despite the importance of the root microbiota, the molecular mechanisms by which plants perceive and respond to commensal bacteria at the level of individual root cell types remain poorly understood.

The exception is the symbiotic interaction between legumes and rhizobia, where specific molecular pathways and genetic components critical for successful establishment are well defined. The common symbiosis signaling pathway (CSSP) in legumes is essential for establishing long-term, intracellular associations with both nitrogen-fixing rhizobia and arbuscular mycorrhizal (AM) fungi[8]. Recognition of Nod- or Myc-factors triggers specific signaling responses, leading to characteristic calcium spiking patterns that are interpreted by CCAMK[9]. These calcium signals subsequently activate *CYCLOPS*, a

transcriptional regulator positioned at the interface between mycorrhizal and rhizobial symbioses[10]. *CYCLOPS*, in turn initiates the transcription of several downstream symbiotic genes, including *NSP1* and *NSP2*, which are essential for nodulation and also promote colonization by AM fungi[11]. In *cyclops* mutants, the early Nod factor perception and Nod factor receptor-mediated signaling events in root hairs, including calcium spiking, remain functional. This allows initial root hair curling and rhizobial entrapment. However, downstream activation of the infection thread (IT) program is impaired, resulting in rhizobia accumulating in infection pockets without successful IT progression[10]. Despite this block in epidermal infection, cortical responses are still activated, leading to initiation of nodule organogenesis in the absence of infection threads[10].

Recent single-cell RNA-sequencing analyses have contributed to elucidation of the cellular expression patterns of symbiotic genes, revealing specific transcriptional profiles within root hair cells that directly interface with rhizobia[12]. Application of similar approaches to the study of root-microbiome interactions is difficult because of the

---

[1]Department of Molecular Biology and Genetics, Aarhus University, Aarhus C, Denmark. [2]These authors contributed equally: Francesca Tedeschi, Johan Quilbé. ✉e-mail: ft@mbg.au.dk; sua@mbg.au.dk

absence of established marker genes for the colonization of the root surface or the entry of commensal bacteria into host tissues. While several studies have examined plant expression responses to mono-inoculation with bacterial strains or within the context of complex microbial communities, these transcriptomic studies predominantly highlighted broad biological processes, including stress responses, immunity, cell wall modification, nutrient uptake, primary metabolism, and epigenetic regulation[13–22]. Here, we use single-cell RNA-sequencing (scRNA-seq) to analyse legume root responses to commensal bacterial communities at cellular resolution, with a focus on transcriptional responses in root hair cells under symbiotic and non-symbiotic conditions.

## Results

### A bacterial SynCom enriched in plant interaction-related genes

To establish our commensal inoculant, we selected 19 fully sequenced strains (SynCom19) from the *Lj*-SPHERE culture collection[18]. We based the strain selection on their content of genes potentially associated with plant interactions and on their rhizosphere colonization characteristics in a diverse panel of *Lotus japonicus* (lotus) accessions[23]. SynCom19 includes strains from eight distinct bacterial families (Supplementary Fig. 1). Many of the strains carry genes associated with plant growth-promoting functions, including cytokinin production (miaABE)[24], ethylene precursor degradation (acdS)[25], phosphate mobilization and solubilization (phyC, gcd, pqqBCDEG)[26], spermidine production (speBE)[27], quercetin degradation (Qdx)[28], antagonism (phlG)[29], and mineral uptake (cysA)[30]. We characterized the taxonomy of the strains using phylogenetic trees based on full-length *recA* sequences and, where possible, classified strains to the species level through whole-genome alignment with reference genomes (Supplementary Fig. 1). SynCom19 does not include compatible symbionts capable of nodulating lotus, allowing us to specifically study plant interactions with a community of commensal bacteria. To verify colonization under the single-cell sequencing conditions, we performed 16S rRNA amplicon sequencing on rhizosphere samples from seven independent plates and on the input inoculum. All SynCom19 strains were recovered from the root-associated communities at 5 days post-inoculation (dpi) (Supplementary Fig. 2). Inoculation with SynCom19 alone did not significantly affect plant biomass, whereas co-inoculation with the symbiont *Mesorhizobium loti* R7A (R7A) and SynCom19 significantly increased biomass relative to mock and Syn-Com19 treatments (Supplementary Fig. 3).

### Pervasive root transcriptional response to commensal bacteria

We previously investigated lotus root responses to its symbiont R7A alone at single-cell resolution[31]. To investigate root cell responses to commensal bacteria only and to commensal bacteria in combination with R7A, we carried out scRNA-seq of protoplasts from lotus wild-type roots 5 dpi with SynCom19, SynCom19 + R7A and mock-inoculated. We characterized a total of 50,996 cells (Supplementary Data 1) with a median of 2101 unique molecular identifiers and ~1200 transcripts per cell after filtering. After integration, the samples were clustered using Seurat[32] (Fig. 1a and Supplementary Fig. 4). Cellular identities of indi-vidual clusters were assigned using homologous markers from Arabi-dopsis, marker gene information from lotus base[33] and promoter-reporter lines as previously reported[31] (Supplementary Data 2).

Using MAST[34] to conduct differential gene expression analysis, we found distinct responses of different cell types to inoculation with SynCom19 at 5 dpi compared to the control (Supplementary Data 3 and Fig. 1c). Most cell types exhibited a higher number of down-regulated than upregulated genes, but root cap and root hair cells showed the opposite trend. The least responsive cell types were xylem, endodermis and phloem. Root cap, root hair, pericycle, stele, and meristem had the highest total number of differentially expressed genes (Fig. 1c).

Expression patterns of all genes can be browsed online using our ShinyApp: https://lotussinglecell.shinyapps.io/lotus_japonicus_single_cell_microbiome/.

To compare the plant root response to SynCom19 in the presence and absence of rhizobia, we first identified genes differentially expressed between the SynCom19 + R7A and mock samples. These included many known to be involved in rhizobium infection, including *NUCLEAR FACTOR Y, SUBUNIT A1* (NF-YA1)[35], *CYSTATHIONINE β-SYNTHASE DOMAIN-CONTAINING PROTEIN 1* (CBS1)[36], *NOD FACTOR RECEPTOR 1* (NFR1) and *NOD FACTOR RECEPTOR 5* (NFR5)[37] (Supplementary Data 4). The infected cortical cells did not form a distinct cluster using Seurat's default dimensionality reduction function (Fig. 1a). In contrast, clustering using TopOmetry[38] produced a distinct cluster (cluster 59, Fig. 1b) containing cortical cells (134 cells in Syn-Com19 + R7A) expressing well-established molecular markers of rhi-zobial infection and early nodule development, such as *NODULE INCEPTION* (NIN)[39], *NODULATION SIGNALING PATHWAY 2* (NSP2)[40], *NODULATION PECTATE LYASE* (NPL)[41], and *EXOPOLYSACCHARIDE RECEPTOR 3* (EPR3)[37] (Fig. 1d, e). These genes were specifically induced in root hairs and cortical cells following inoculation with Syn-Com19 + R7A (Fig. 1d, e, and Supplementary Data 5). TopOmetry cluster 59 was also represented in the mock (38 cells) and SynCom19 samples (187 cells). However, cells from these samples did not exhibit expression of genes associated with rhizobial infection or nodulation (Fig. 1d, Supplementary Data 6 and 7). The lack of expression of molecular markers of rhizobial infection in the mock and SynCom19 samples indicates that there was no R7A cross-contamination and that Nod-factor signaling was exclusively active in the SynCom19 + R7A samples.

### Overlapping root hair transcriptional responses to symbionts and commensals

Having confirmed the lack of canonical rhizobium Nod-factor signaling in SynCom19 only inoculated samples, we proceeded to investigate if there were other overlaps in the transcriptional responses to symbiotic and commensal bacteria. First, we compared differentially expressed genes in cell clusters for our SynCom19 treatment and the response to R7A alone in a previously published dataset[31] (Supplementary Data 2 and 8 We found a large overlap of more than 900 genes, indi-cating overlapping responses (Supplementary Fig. 5a), but these were broadly distributed across clusters and not necessarily induced in the same cell types. To specifically assess whether transcriptional responses to symbiotic and commensal bacteria overlap in cell types associated with infection processes, such as root hair infection threads and nodule cells, we assessed the overlap between genes upregulated by SynCom19 at 5 dpi and the R7A-associated marker genes identified in a previous study[31]. Specifically, SynCom19-induced genes were compared with markers of infected and nodule cells identified at 10 dpi (Supplementary Data 9 and 10), applying thresholds of adjusted $p ≤ 0.05$ and pct.2 ≤ 0.02. This analysis identified 17 genes commonly induced (Supplementary Data 11 and Supplementary Fig. 5b). Among these genes, *LotjaGi5g1v0322700* exhibited a distinct induction pat-tern in a subset of rhizobium-responsive root hairs (Fig. 2a, b). *Lotja-Gi5g1v0322700* encodes a bHLH transcription factor, which is a close homolog of the Arabidopsis *AT1G27740* gene known as *ROOT HAIR DEFECTIVE 6-LIKE 4* (RSL4), and we named it *ROOT HAIR DEFECTIVE 6 LIKE A* (RHD6LA).

In addition, we integrated our current data, wild-type lotus roots treated with SynCom19 with and without R7A at 5 dpi, with previously published scRNA-seq data from wild-type lotus roots 5 and 10 dpi with R7A and lotus *cyclops* mutant roots 5 dpi with R7A[31] to allow direct comparison of cell populations across studies (Fig. 2c).

The SynCom19 commensals do not activate Nod factor signaling, whereas R7A does (Figs. 1d and 2d). In contrast, both treatments induce *RHD6LA* in root hairs (Fig. 2a, b, e). To further investigate the

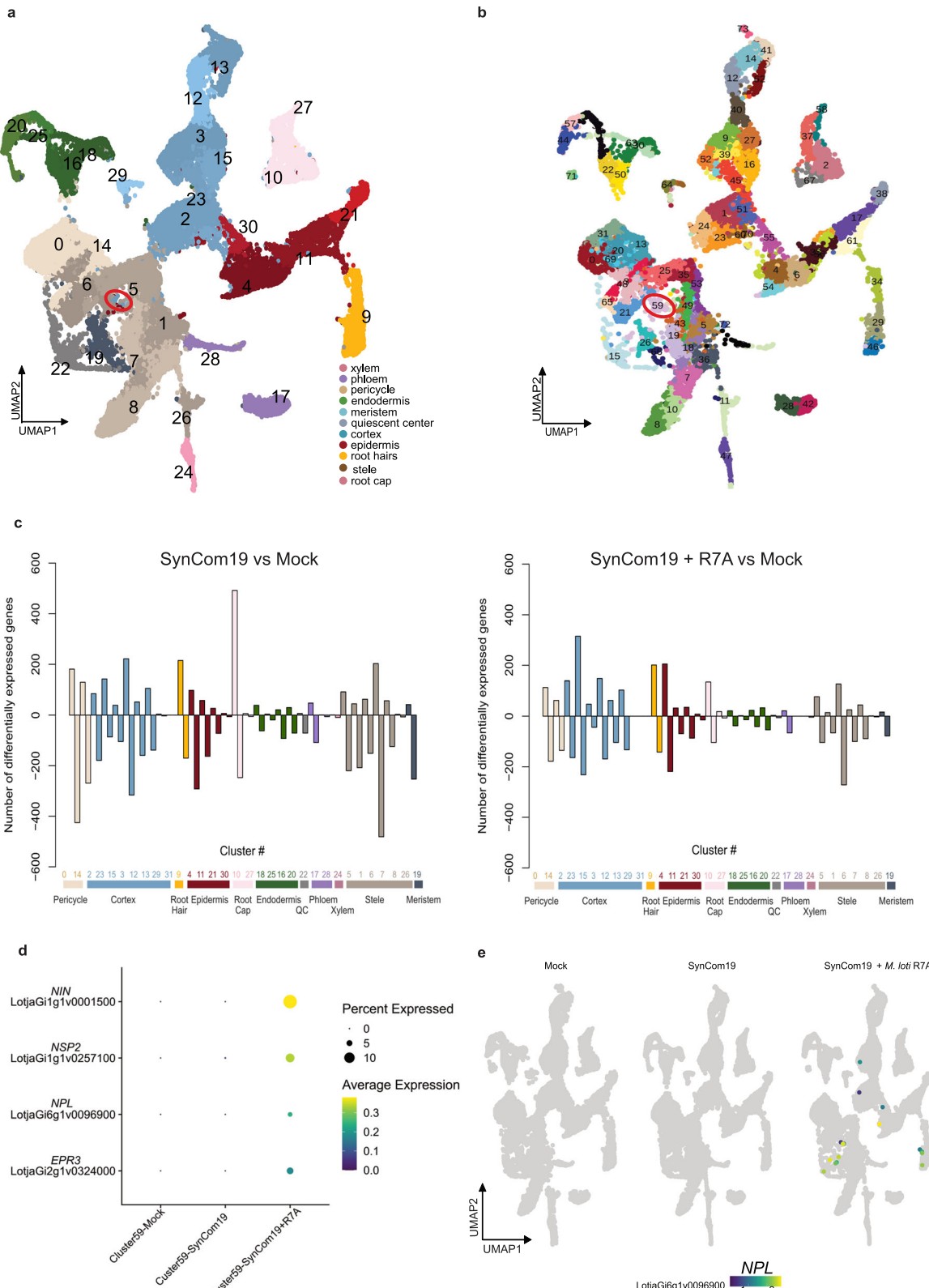

**Fig. 1 | A cellular atlas of lotus roots inoculated with SynCom19 ± *M. loti* R7A.**
**a** UMAP of control, SynCom and SynCom + *M. loti* R7A inoculated root cells 5dpi showing clusters for known root cell types. The red circle shows the area of cortical cells that did not form a distinct cluster using Seurat's default dimensionality reduction function. **b** UMAP re-clustering using the cluster numbers created with TopOMetry[38] allowed assignment of a cortical cell cluster (cluster 59, red circle) expressing marker genes for rhizobium infection. **c** Differentially expressed genes across clusters in response to SynCom19 compared to mock (left) and for

SynCom19 + *M. loti* R7A compared to mock (right). Complete lists of differentially expressed genes and marker genes can be found in the Supplementary data. **d** Dot plot showing the average expression (color scale) and proportion of cells expressing each gene (dot size) in TopOMetry cluster 59 across samples. The four genes are well-known infection-related markers and characterize cluster 59 in the Syn-Com19 + R7A sample only. **e** Example of a symbiosis-specific gene (*NODULATION PECTATE LYASE, NPL*) induced by *M. loti* R7A.

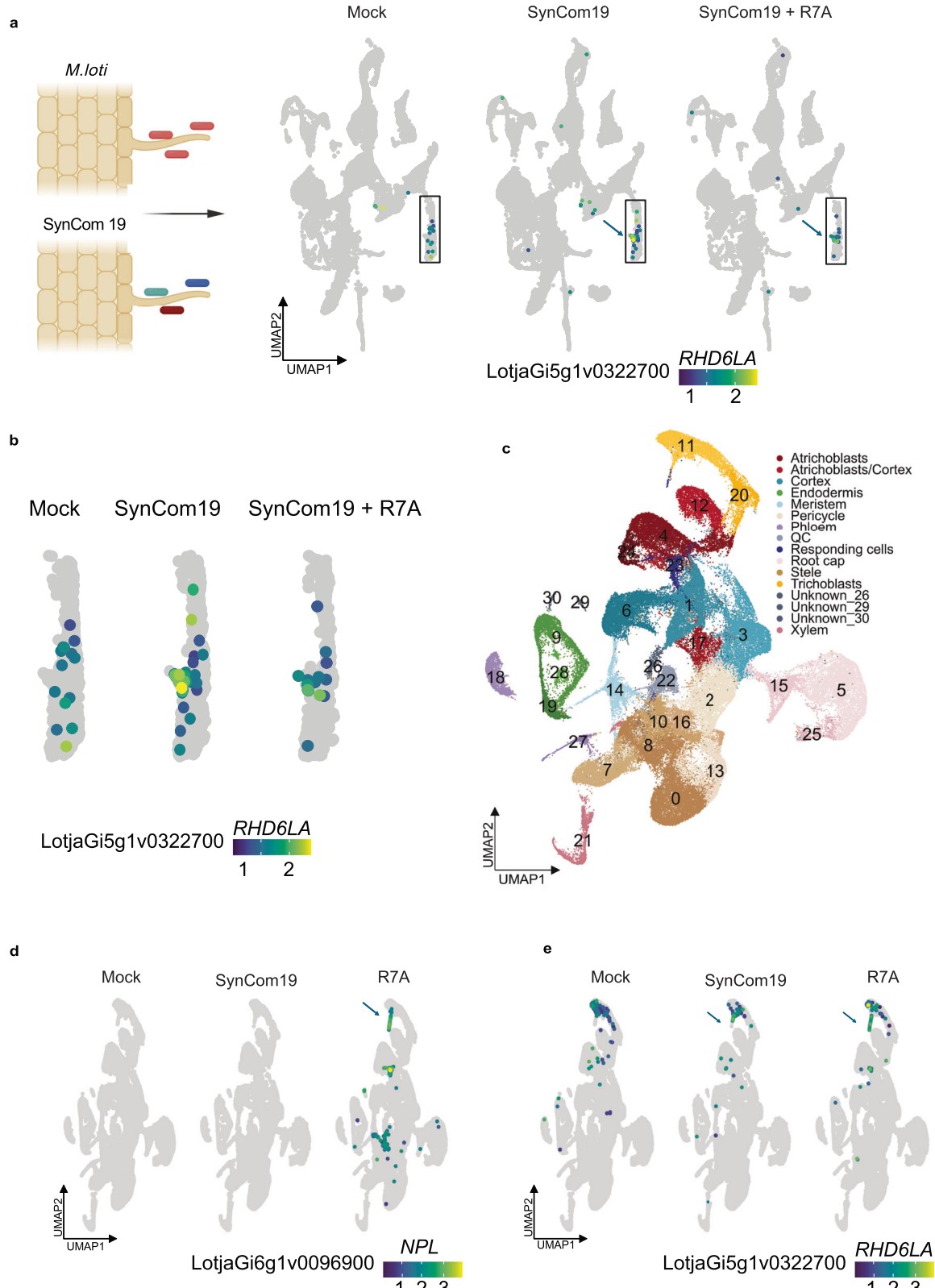

**Fig. 2 | _RHD6LA_ is induced in root hairs upon bacteria inoculation. a** The boxed region highlights the root hair population, with blue arrows indicating cells showing _ROOT HAIR DEFECTIVE 6 LIKE A_ (_RHD6LA_) induction in response to both SynCom19 and SynCom19 + R7A treatments. Created in BioRender. Tedeschi, F. (2026) https://BioRender.com/6jjg7se. **b** Zoom of the subset of root hair cells exhibiting _RHD6LA_ induction in response to both SynCom19 and SynCom19 + R7A. **c**–**e** The current SynCom19 data integrated with previously published datasets[31]. **c** Cell type annotations. **d** Expression of the rhizobium infection marker _NODU-LATION PECTATE LYASE_ (_NPL_). **e** Induction of _RHD6LA_ in a subset of root hairs observed only in samples inoculated with SynCom19 and R7A.

root hair response to commensals exemplified by *RHD6LA*, we re-clustered mock and SynCom19 root hair cells, resulting in ten sub-clusters (Fig. 3a). Among these, only subcluster 7, comprising 92 out of 1658 root hair cells, consistently expressed *RHD6LA* in the SynCom19-treated samples (Fig. 3b). This subcluster represents the SynCom19-responsive root hair cells, and we will refer to it as SynCom19_RH-subcluster7 (Fig. 3b).

## Wild-type SynCom19 responses resemble those of *cyclops* to R7A

Our second integrated object (Fig. 2c) comprises data from our current study and a previously published study[31]. The integrated samples include mock, SynCom19, SynCom19 + R7A and R7A-treated wild-type plants as well as mock and R7A-treated *cyclops*[10] mutants. Focusing on *RHD6LA* expression, we observed pronounced *RHD6LA* induction by R7A in *cyclops* root hairs (Fig. 3c). Since Nod factor signaling is absent in SynCom19 and only partly active in *cyclops*[10] mutants, we asked if the wild-type SynCom19-responsive root hair cells transcriptionally more closely resemble R7A-responsive root hairs in wild-type or in the *cyclops*[10] mutant. To test this, we compared the transcriptional profiles of the three root hair populations: SynCom19_RHsubcluster7 (SynCom19-responsive root hairs at 5 dpi, 44 cells, identified in this study), Infected_RH_5dpi (R7A-responsive root hairs at 5 dpi, 13 cells, defined in a previous study[31]), and RH_cyclops_5 (*cyclops* mutant root hairs at 5 dpi showing an aberrant response to R7A, 149 cells, defined in a previous study[31]) (Fig. 3d).

To quantify the transcriptional similarity between the three root hair populations, we compared the expression of all genes across the three subpopulations. We found a stronger correlation between Syn-Com19_RHsubcluster7 and RH_cyclops_5 than between these two populations and Infected_RH_5dpi (Supplementary Fig. 6). To determine if this was also the case for the genes that were specifically expressed in each of the three subpopulations, we identified these by comparing the three populations to the corresponding mock-inoculated root hair cells at 5 dpi. This approach identified 205 marker genes for SynCom19_RHsubcluster7, 159 for infected_RH_5dpi, and 189 for RH_cyclops_5 cells, totaling 466 unique genes ($p$-adj $\leq 0.05$, pct.1 > 0.1, pct.2 ≤ 0.1) (Fig. 3e, Supplementary Data 12, 13, and 14). Subsequently, we examined the expression levels of the 466 marker across populations, confirming a significantly higher correlation between SynCom19_RHsubcluster7 and RH_cyclops_5 root hair expression profiles than between SynCom19_RHsubcluster7 and Infected_RH_5dpi cell ($P < 0.0001$, Pearson and Filon's z (1898) test) (Fig. 3f), a pattern also observed at the level of individual gene expression (Supplementary Fig. 7). It is worth noting that the correlation at the transcriptional level between RH_cyclops_5 and Syn-Com19_RHsubcluster7 cells was higher than that between RH_cyclops_5 and Infected_RH_5dpi ($P = 0.0002$, Pearson and Filon's z (1898) test), although both these populations were inoculated with R7A and were assayed in the same experiment. SynCom19 thus induces a transcriptional state in wild-type root hair cells, which resembles that of R7A-treated *cyclops* mutants characterized by activation of genes associated with early root hair responses in the absence of infection thread progression[10].

## *RHD6LA* controls root hair response to commensals and symbionts

We identified 16 overlapping genes between the SynCom19_RH-subcluster7, infected_RH_5dpi and RH_*cyclops*_5 markers (Fig. 3e and Supplementary Data 15). In addition to induction during a compatible symbiotic interaction, these were all induced in the absence of Nod factor by SynCom19 and by R7A in the absence of the CSSP transcription factor CYCLOPS, suggesting that they may be components of a general bacterial perception system in root hairs. *RHD6LA* was included in this set of 16 genes, but its function has not been described.

Since *RHD6LA* was induced by both symbionts and commensals, it could potentially play a role in both interactions. Legume roots respond to the presence of symbiotic rhizobia through the curling of root hairs to facilitate bacterial attachment and subsequent infection in a Nod factor-dependent manner[42]. Given that the same population of root hairs exhibited a transcriptional response to SynCom19 (Fig. 2d), we investigated if SynCom19 also elicits a morphological response in root hairs. To assess this and explore the potential involvement of *RHD6LA*, we obtained two *LORE1 rhd6la* insertion lines[43] and examined root hair morphology in wild-type and *rhd6la* mutant plants at 5 dpi with SynCom19 using confocal microscopy. Without inoculation, the *rhd6la* mutants showed root hair morphology indistinguishable from the wild type. With SynCom19 and R7A treatment, most root hairs did not respond, but displayed a normal elongated and tubular shape (Fig. 4c, e). Some wild-type and *rhd6la* mutant root hairs, however, exhibited clear deformation upon inoculation with SynCom19 (Fig. 4d, g, h). The swollen and other abnormal root hair phenotypes observed in the presence of commensals were significantly more frequent in the *rhd6la* mutants, indicating that *RHD6LA* plays a role in regulating root hair responses to commensal bacteria (Fig. 4i). To determine if *RHD6LA* also plays a role in symbiotic interactions, we compared wild type and *rhd6la* infection thread (10 dpi) and nodule (21 dpi) counts. Consistent with a compromised symbiotic response, comparison with the wild type revealed a significant reduction in *rhd6la-1* and *rhd6la-2* for infection thread density and nodule count (Fig. 4j, k). Because *RHD6LA* expression is confined to root hairs and absent from nodules or primordia, these findings suggest that the lower nodule numbers observed in the *rhd6la* mutants are a consequence of reduced infection thread formation, rather than a result of a direct role of *RHD6LA* in nodule organogenesis. In line with its transcriptional profile, *RHD6LA* thus regulates root hair responses to both commensal and symbiotic bacteria. To determine whether *rhd6la* broadly regulates root hair responses to commensal bacteria, and to exclude the possibility that the observed phenotype is driven by a single strain, we repeated the experiment using four individual strains (LjR62, LjR82, LjR281, and LjR296), each representing a distinct bacterial family (*Comamonadaceae*, Flavobacteriaceae, Pseudomonadaceae, and Burkholderiaceae), in mono-inoculation rather than the original 19-member synthetic community. Three of the four tested strains induced a higher frequency of aberrant root hair phenotypes in the *rhd6la* mutant compared with the wild type (Fig. 4l–t).

## Discussion

scRNA-seq studies have provided significant insights into the transcriptional regulation of symbiotic signaling in legume-rhizobia interactions, highlighting the complexity and specificity of gene expression in different cell types and stages of symbiosis[12]. Commensals are known to interact with root systems and potentially influence plant health and growth[18,19]. Recent single nucleus transcriptomic studies have begun to resolve cell-type-specific root responses to individual beneficial or pathogenic microbes[13]. However, how roots transcriptionally respond to more complex commensal communities, and how these responses are related to symbiosis-associated transcriptional programs, has so far remained largely unexplored.

Our study demonstrates that plant roots mount a substantial transcriptional response to commensal bacteria. It differs between cell types and includes a specific response in a population of rhizobium-responsive root hairs. The transcription factor CYCLOPS is well known for its essential role in intracellular infection of plant roots by rhizobia and arbuscular mycorrhizal fungi[10,44]. In *cyclops* mutants, Nod factor perception remains intact, allowing root hair curling and initiation of nodule primordia, but rhizobial infection is abrogated at an early stage, leaving rhizobia trapped in curled root hairs[10] (Fig. 5). We observed that commensal bacteria, which do not produce Nod factors, induce a

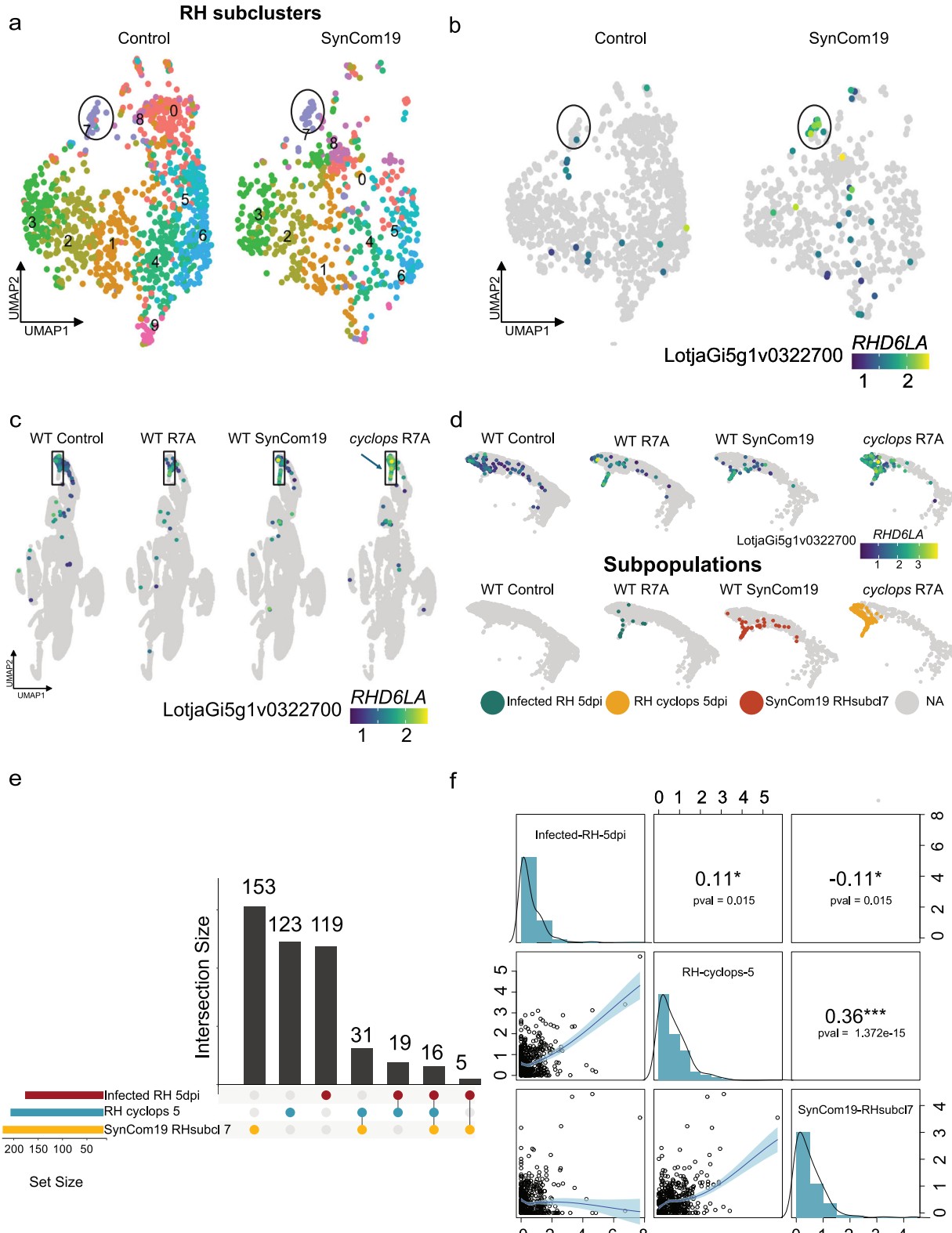

**Fig. 3 | Similarities between wild-type SynCom19 and *cyclops* R7A responses.**
**a** UMAP of reclustered root hair cells from control and SynCom19 samples.
**a**, **b** Subcluster 7 (black outline) showed strong and specific induction of *ROOT HAIR DEFECTIVE 6 LIKE A* (*RHD6LA*) on SynCom19 inoculation. **c** A specific root hair population shows a high induction of *RHD6LA* in *cyclops* upon R7A inoculation. **d** A subset of the root hair cells in the integrated object. The top panel shows the expression of *RHD6LA*. The bottom panel shows the three cell subpopulations used for the correlation analysis. **e** Upset plot showing the markers for RH_*cyclops*, SynCom19_RH_subcluster7 and infected_RH_5 dpi markers[31] and their overlaps (pct.1 ≥ 0.1, pct.2 ≤ 0.1). **f** Bivariate scatter plots, histograms, and Spearman correlation values for the log2 average expression of 466 markers in the three indicated populations. The blue line in the scatter plots denotes LOWESS smooth with 95% confidence intervals. The *p* values for the correlations were calculated using *t* approximation.

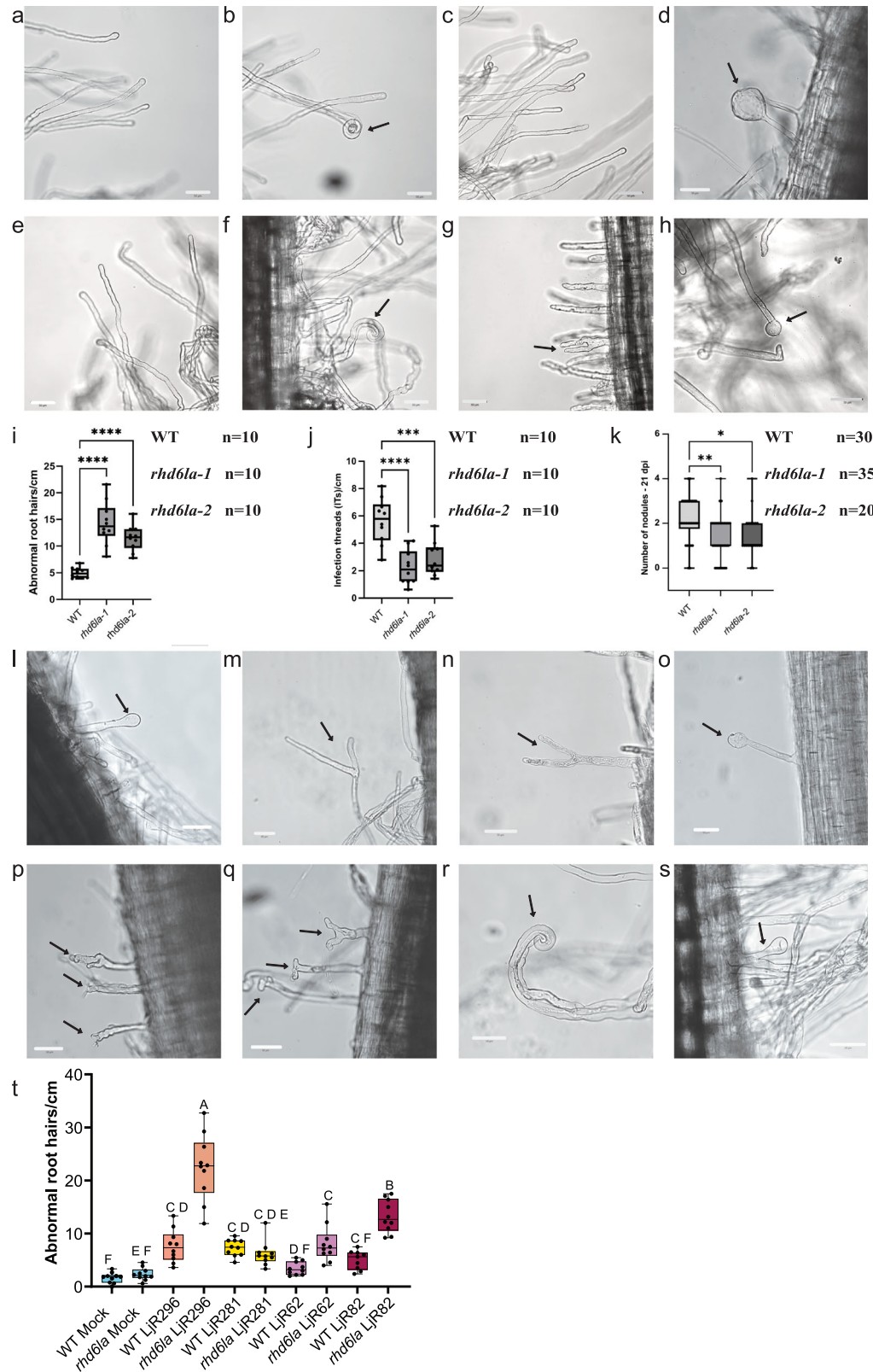

transcriptional response in wild-type root hairs that overlaps significantly with the response to Nod factor-producing rhizobia in *cyclops* mutant root hairs (Fig. 3e, f). This similarity likely reflects a shared signaling state in which root hairs perceive bacteria but do not progress to infection thread formation. In this context, SynCom19-treated wild-type root hairs and R7A-treated *cyclops* root hairs both activate early bacterial perception programs in the absence of

downstream symbiotic signaling mediated by the CSSP/CYCLOPS module. By contrast, wild-type root hairs inoculated with R7A engage the full symbiotic transcriptional program downstream of CSSP/CYCLOPS, resulting in a response that diverges from that elicited by commensal bacteria.

This transcriptional overlap could, in principle, be caused by a general stress response in root hair cells that experience incompatible

**Fig. 4 | Responses to inoculation at 5 dpi (*M. loti* R7A, SynCom19 and individual commensal strains) in wild-type and *rhd6la* root hairs.** **a–h** Confocal pictures of root hair responses. **a** Wild-type (WT) nonresponsive root hair cells. **b** WT root hair curling in response to R7A inoculation, **c** WT nonresponsive root hairs inoculated with SynCom19, **d** WT root hair responding to SynCom19 (arrow indicates a swollen root hair tip), **e** *rhd6la* nonresponsive root hairs after inoculation with either R7A or SynCom19, **f** *rhd6la* root hair curling in response to R7A (arrow indicate a curled root hair), **g** *rhd6la* root hair responding abnormally to SynCom19 (arrow indicates a bifurcated root hair), **h** *rhd6la* root hair responding to SynCom19 (arrow indicates a swollen root hair). **i** Ten entire roots were examined, and wild-type and mutant root hairs were scored for abnormal phenotypes at 5 dpi. Statistical analysis was performed using one-way ANOVA followed by Dunnett's multiple comparisons test (two-sided) to compare *rhd6la-1* and *rhd6la-2* to the WT control. *P* values were adjusted for multiple comparisons. ****$P < 0.0001$. **j** Infection thread (IT) counts in wild-type ($n = 10$) and *rhd6la* mutant ($n = 10$) plants 10 dpi. Statistical significance was determined by one-way ANOVA followed by Dunnett's multiple comparisons test (two-sided), comparing each mutant to WT. *P* values were adjusted for multiple

comparisons ***$P < 0.001$, ****$P < 0.0001$. **k** Nodulation assay of wild-type ($n = 30$) and mutant plants (*rhd6la*-1 $n = 35$; *rhd6la-2* $n = 20$) inoculated with R7A 21 dpi. Statistical significance was determined by one-way ANOVA followed by Dunnett's multiple comparisons test (two-sided), comparing each mutant to WT. *P* values were adjusted for multiple comparisons. *$P = 0.0358$; **$P = 0.0037$. Representative aberrant phenotypes in wild-type (Gifu) root hairs inoculated with **l** LjR62, *Comamonadaceae* **m** LjR82, Flavobacteriaceae **n** LjR281, Pseudomonadaceae, **o** LjR296. Burkholdariaceae. Representative phenotypes in *rhd6la* mutant root hairs inoculated with **p** LjR62, **q** LjR82, **r** LjR281, **s** LjR296. Boxplot in **t** shows abnormal root hairs per cm for the indicated genotypes and treatments. Ten entire roots were examined, and wild-type and mutant root hairs were scored for abnormal phenotypes at 5 dpi. Statistical differences among groups were assessed by one-way ANOVA followed by Tukey's HSD post hoc test. For boxplots represented in **i–k, t,** center lines indicate the median, boxes the interquartile range, and whiskers the minimum and maximum values; points represent individual biological replicates. Bars, 50 μM.

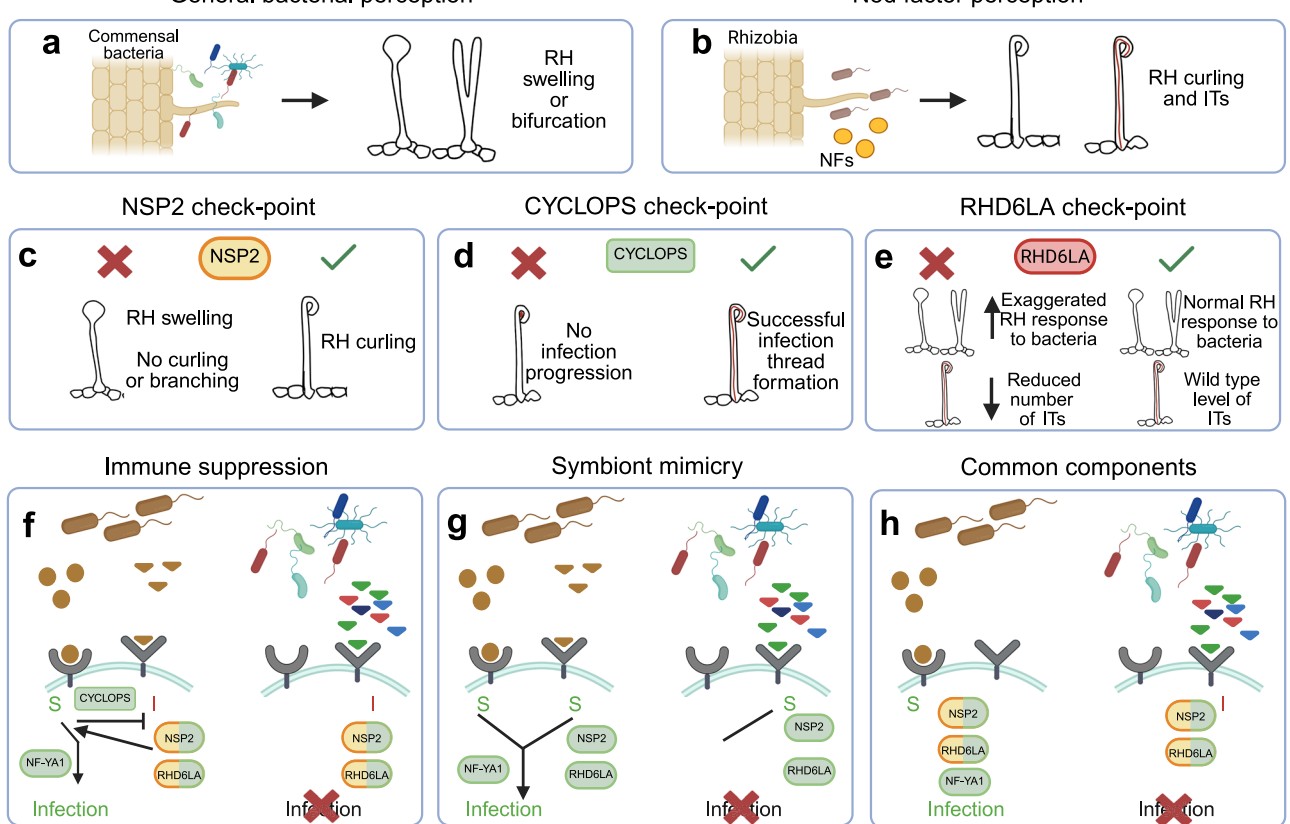

**Fig. 5 | Scenarios for root hair responses to commensal and symbiotic bacteria.** **a** Root hair perception of commensal bacteria, which do not produce Nod factors, lead to swelling or bifurcation, but not to infection (Fig. 4d, g, h). **b** Nod factor-producing rhizobia induce root hair curling and infection thread formation. **c** *nsp2* mutants respond to symbiotic rhizobia only with root hair swelling. No curling or infection thread formation is observed[53]. **d** Root hair curling is observed in *cyclops* mutants, but no infection threads are formed[10]. **e** *rhd6la* mutants show exaggerated root hair deformation in response to commensal bacteria and a reduced number of infection threads on rhizobium inoculation (Fig. 4i–k). **f** Immune suppression scenario. Symbiotic rhizobia induce both symbiotic (S) and immune (I) signaling.

Symbiotic signaling suppresses immune signaling in a CYCLOPS-dependent manner. RHD6LA and NSP2 promote symbiont infection in combination with Nod factor signaling. Commensal bacteria only induce immune signaling. **g** Symbiont mimicry scenario. The non-Nod factor signal is part of the symbiotic response. Commensal bacteria mimic the non-Nod factor signal, but the lack of Nod factors prevents infection. **h** Common components scenario. Symbiotic rhizobia do not necessarily induce immune signaling, but NSP2 and RHD6LA are downstream of both Nod factor and Nod factor-independent bacterial perception. Created in BioRender. Tedeschi, F. (2026) https://BioRender.com/hklwa7j.

microbial interactions, or competition for nutrients, regardless of whether they result from challenge by nonsymbiotic commensals or deficient CSSP signaling during rhizobium infection. If the overlap represented a general stress response, it would be expected to occur broadly across root hair cells exposed to commensals. However, the specificity of the transcriptional responses to rhizobium-responsive

root hair argues against this hypothesis. Instead, it is likely that a general, Nod factor-independent, bacterial perception system exists in rhizobium-responsive root hairs alongside the specific Nod factor perception mediated by Nod factor receptors (Fig. 5f–h). Here, we explore different scenarios for how the observed transcriptional overlap could be generated.

Activation of the general bacterial perception system alone would not promote infection and could represent an immune signal, but the general system would also induce components, including RHD6LA and NSP2, that promote rhizobium infection in combination with Nod factor signaling (Fig. 5d). The fact that infection fails in *cyclops* rhizobium-responsive root hairs, despite initiation of Nod factor signaling, suggest a scenario where CYCLOPS could play a role in suppressing immune signaling (Fig. 5f).

An alternative scenario is that the Nod factor-independent bacterial perception system could be an integral part of rhizobium recognition that has been hijacked by certain commensals, enabling them to activate part of the symbiotic program in responsive root hairs (Fig. 5g). By tagging along with Nod factor-producing rhizobia, this could allow commensals to gain access via intracellular infection to root nodules that make up attractive, nutrient-rich ecological niches. In this scenario, CYCLOPS would not be required for suppression of immunity, but merely for full activation of the symbiotic program to complement the Nod factor-independent signal.

A third scenario is that the transcriptional overlap is caused by common signaling components used downstream of both general bacterial and Nod factor perception (Fig. 5h). The transcriptional overlap between *cyclops* + R7A and wild-type + SynCom19 would then be due to Nod factor and general bacterial perception signals feeding into the same signalling pathway, which is normally pushed towards infection promotion by compatible rhizobia through CYCLOPS-dependent CSSP signalling.

Root hair morphological responses offer some support for the third scenario (Fig. 5h). *nsp2* mutants are able to perceive Nod factors and initiate calcium spiking, but do not show root hair curling. Instead, *nsp2* root hairs exhibit swelling and deformation reminiscent of the wild-type root hair response to commensals detailed here (Figs. 4g, h, and 5a, c). This morphological root hair response to commensals is exaggerated in *rhd6la* mutants, which also show reduced infection thread and nodule counts (Figs. 4j, k, and 5e). In contrast, Nod factor-deficient rhizobia do not induce root hair deformation[45] So, whereas Nod factors are needed for root hair deformation in response to rhizobia, commensals can induce root hair deformation in the absence of Nod factor signaling, just like they can induce *NSP2* and *RHD6LA* expression independent of Nod factors.

The scenarios presented in Fig. 5 are simplifications. In reality, Nod factors, conserved microbe-associated molecular patterns, microbial effectors and plant detection systems result in a great variation in interaction outcomes, especially when no fully compatible symbionts are available[46]. This extends also to commensals co-colonizing nodules with compatible rhizobia[47]. However, determining the reasons underlying the overlapping root hair transcriptional responses to commensals and symbionts could help us take the next steps in understanding how legumes balance the benefits of symbiosis with the risk of exploitation by potential pathogens.

From an evolutionary perspective, the most parsimonious explanation for our findings is that the general bacterial perception system in root hairs predates symbiosis-specific Nod factor perception, which would be a later adaptation added in the process of enabling bacterial endosymbiosis. However, the localization of the transcriptional response to a subset of root hairs involved in rhizobium signalling would suggest that the general perception system in responsive root hairs could be a legume-specific innovation serving to stabilize rhizobium endosymbiosis by providing protection against cheater bacteria. The phylogenomic distribution of *RHD6LA* is compatible with this hypothesis, since *RHD6LA* appears to result from a legume-specific duplication occurring around the time intracellular rhizobium infection evolved and show signs of pseudogenization in peanut, which does not produce infection threads in root hairs (Supplementary Fig. 8).

In conclusion, our study reveals a previously unappreciated complexity in plant-microbe interactions at the root hair interface (Fig. 5). The interplay between general bacterial perception and symbiosis-specific Nod factor signaling in select root hairs provides a new framework for investigating the molecular mechanisms underlying plant-microbe recognition and the evolution of symbiotic relationships.

## Methods

### Plant material

*Lotus japonicus* seeds from the Gifu accession (both wild type and mutants) were subjected to scarification using sandpaper, followed by sterilization using a 1% (v/v) sodium hypochlorite solution for 8–10 min. Subsequently, the seeds underwent five washes with sterile water in sterile conditions. After being kept at 4 °C overnight, the seeds were moved to square Petri dishes for germination under a 16-h day cycle (at 21 °C) followed by an 8-h night cycle (at 19 °C). After 4 days, the seeds with emerging radicles were moved to square plates containing 1.4% Agar Noble slopes supplemented with 0.25× B&D medium. These plates were covered with filter paper. A metal bar featuring 3-mm holes for roots was introduced at the upper edge of the agar slope. Plant growth plates containing 10 seedlings each were then inoculated with 500 μL of microbial communities suspended in water with an $OD_{600} = 0.02$ and applied along the length of the roots. For genetic studies, the *LORE1* lines 30107507 (*rhd6la-1*) and 30035826 (*rhd6la-2*) were used[43]. A *LORE1* line is a lotus plant line in which the endogenous *LORE1* retrotransposon has been inserted into a specific gene, creating a stable insertional mutant[43].

### Bacterial strains

The 19 bacterial strains (SynCom19) were inoculated in tryptic soy broth (TSB) media for 2 days at 28 °C, then centrifuged at $4000 \times g$ for 15 min. The supernatant was discarded, and the pellet was resuspended in 1 mL water. The $OD_{600}$ was measured, and the bacterial suspensions were diluted to obtain an $OD_{600}$ equal to 0.02. The *M. loti* R7A rhizobia strain was cultured for 2 days at 28 °C in yeast mannitol broth (YMB) agar plate, resuspended in water and diluted accordingly to obtain an $OD_{600}$ equal to 0.02.

### Protoplast isolation and scRNA-seq

For the isolation of protoplasts, whole roots were subjected to protoplasting under gentle agitation over a 3-h period at room temperature using a 5 mL digestion solution. This digestion solution contained 10 mM MES (pH 5.7), 1.5% (w/v) cellulase R-10, 2% (w/v) macerozyme R-10, 0.4 M D-sorbitol, 10 mM $CaCl_2$, 5% (v/v) viscozyme, and 1% (w/v) BSA. The resultant intact protoplasts were separated by straining the protoplast-containing digestion solution through a 40 μM strainer into 15 mL Falcon tubes. This mixture was then combined with 5 mL of a 50% Optiprep solution, composed of 50% (v/v) Optiprep, 10 mM MES (pH 5.7), 0.4 M D-sorbitol, 5 mM KCl, and 10 mM $CaCl_2$. To the combined solution, 2 mL of a 12.5% Optiprep solution (12.5% (v/v) Optiprep, 10 mM MES (pH 5.7), 0.4 M D-sorbitol, 5 mM KCl, 10 mM $CaCl_2$) and 250 μL of a 0% Optiprep solution (10 mM MES (pH 5.7), 0.4 M D-sorbitol, 5 mM KCl, 10 mM $CaCl_2$) were cautiously added in sequence. Subsequent to this, the tubes were centrifuged at $250 \times g$ for 10 min at 4 °C. Living protoplasts were collected at the interface between the 12.5 and 0% Optiprep solutions and counted using a Neubauer chamber. For the generation of single-cell RNA sequencing libraries, the Chromium Next GEM Single Cell 3′ Kit v3.1 (10X Genomics) was utilized in accordance with the manufacturer's instructions, with the aim of recovering 5000 cells for each biological replicate.

### Microbiome profiling

For rhizosphere sampling, the paper substrate surrounding the roots of 10 plants per plate was collected and pooled. The material was transferred into a 50 mL Falcon tube containing sterile water,

vortexed, and 300 μL of the resulting suspension was transferred to a 2 mL microcentrifuge tube for DNA extraction using the FastDNA Spin Kit for Soil (MP Biomedical). As a control, an aliquot (300 μL) of the SynCom input culture was collected prior to inoculation and stored in a 2 mL tube for DNA extraction. The DNA samples were used to construct 16S amplicon libraries. PCR amplification of the V5–V7 region was performed with Phusion polymerase (ThermoFisher) and primers Read1-PCR1-799F/Read2-PCR1-1193R (Supplementary Data 16) with flanking regions under the following conditions: 98 °C 30 s; 25 cycles of 98 °C 10 s, 60 °C 10 s, 72 °C 5 s; final extension 72 °C 5 min. Products were diluted 1:3 and re-amplified (10 cycles) with Nextera indexing primers. Amplicons were purified with magnetic beads, quantified by PicoGreen (LightCycler 480 II, Roche), pooled to 20 ng/sample and concentrated by a second bead purification. Final library concentration was measured with Qubit 2.0 (Invitrogen) and stored at −20 °C before sequencing on a NovaSeq5000 platform (2 × 250 bp, 20 Gb; Novogene). Sequencing data were processed with QIIME2 to generate ASV tables, with taxonomy assigned by VSEARCH (97% identity) against a custom 16S database. Downstream analysis was conducted in R using the Phyloseq package. ASVs with <1000 total reads were discarded, data were rarefied, and relative abundance calculated to produce stacked barplots.

### Raw data pre-processing, integration and clustering

The initial sequencing data were processed using Cell Ranger v6.1.2 from 10X Genomics (Cell Ranger count summary available in Supplementary Data 17). The genome assembly and gene annotations were established using Lotus Gifu v1.2 and Gifu v1.3 as references, respectively. These references are accessible via Lotus Base[33]. The aligner used for the analysis was STAR v2.7.2a[48]. Employing the default parameters, "cellranger count" was executed. The subsequent stages utilized the "filtered_feature_bc_matrix" obtained from this process as input. Subsequent analyses were conducted using Seurat version 4.0.5[49]. Specifically, cells with an expression of fewer than 200 or more than 7500 genes, along with fewer than 500 UMIs, were removed from the dataset. Further filtering was performed based on the expression of genes associated with mitochondrial and chloroplast genomes. Cells expressing less than 5% of their total read counts were retained after this filtering step. Normalization across all samples was achieved using the "sctransform" function embedded within Seurat, where the "vars.to.regress" parameter was set to encompass mitochondrial and chloroplast genes. Subsequently, the samples were integrated through Seurat's canonical correlation analysis integration pipeline. Employing the integrated data assay, a principal component analysis (PCA) was performed using Seurat's default function for dimensionality reduction. Following this, the "FindNeighbors" and "RunUMAP" functions were applied, utilizing 50 principal components for the entire datasets and 30 for the subsets related to root hairs. Cells were clustered using an unsupervised Louvain clustering algorithm with the default resolution of 0.8. For the integrated object containing data from the current study and 5 and 10 dpi *M. loti* R7A data[32], analysis was performed using Seurat 5.1.0. The same analysis procedure as described above was followed, except for the Louvain clustering resolution, where a value of 0.5 was used. The topometry analysis[38] was performed on the scaled expressions for the 3000 genes in the integrated assay of the Seurat object. The analysis was executed using the default command with the kernel set to "bw_adaptive", the eigen methods used were "msDM", "DM"and the "MAP" projection. The clustering resolution was set to 2.

ShinyCell was used to create the web interface for browsing the single-cell datasets (https://lotussinglecell.shinyapps.io/lotus_japonicus_single_cell_microbiome/)[50].

### Differential gene expression

Differential gene expression analysis between treatments for each cluster and the markers for specific groups of cells were identified using the "MAST" algorithm v 1.16 61 with the "min.pct" = 0.01 parameter. Genes satisfying the criteria of an adjusted $p$-value of ≤ 0.05 and a log fold change greater than 0.25 were classified as differentially expressed.

### Correlation analysis

We identified differentially expressed genes in the three cell populations (Infected_RH_5dpi, SynCom19_RHsubcluster7, and RH_cyclops) compared to the root hairs for their corresponding mock sample using the default method in Seurat. Genes were filtered for pct.1 ≥ 0.1, pct.2 ≤ 0.1, and adjusted $p$ ≤ 0.05. We computed log2 average pseudobulk counts for all the differentially expressed genes across the three cell populations. The correlogram was constructed using the psych R package[51]. The significance between correlation values was calculated using the cocor package with the Pearson and Filon's (1898) z test[52].

### Confocal microscopy

Confocal microscopy was performed with Zeiss LSM780 microscope.

### *RHD6LA* phylogenetic tree

Protein sequences were retrieved from Phytozome 13 (the plant genomics resource). Protein sequences have been aligned with MUSCLE software, and a maximum likelihood phylogenetic tree has been generated using 500 bootstrap replicates for node support values (WAG model, gamma distributed, partial deletion 90).

### Reporting summary

Further information on research design is available in the Nature Portfolio Reporting Summary linked to this article.

## Data availability

The sequencing data generated in this study have been deposited as ENA project accession PRJEB89066. UMAPs and gene expression data can be browsed at [https://lotussinglecell.shinyapps.io/lotus_japonicus_single_cell_microbiome/]. All generated gene lists are provided as Supplementary data. The RDS files have been uploaded to FigShare [https://doi.org/10.6084/m9.figshare.30271879]. Source data are provided with this paper.

## Code availability

The code used for single cell analysis, subclustering, 16S amplicon sequencing, DEGs, correlation analysis and GO term enrichment analysis can be found in Zenodo [https://doi.org/10.5281/zenodo.18598525].

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

## Acknowledgements

This work was supported by the Novo Nordisk Foundation grant [no. 307NNF19SA0059362] for the InRoot project coordinated by Jens Stougaard and by Independent Research Fund Denmark [no. 1026-00032B] to S.U.A.

## Author contributions

F.T., J.Q., and S.U.A. conceived and designed experiments; F.T., J.Q. conducted experiments; F.T., L.I.F. performed the bioinformatic scRNAseq analysis, J.Q. analyzed the 16S amplicon sequencing results; L.I.F. developed the Shiny app, F.T., J.Q., and S.U.A. analyzed the data; S.J.V.C. genotyped the mutant alleles; J.Q., F.T. performed microscopy; F.T. drafted the first version of the manuscript; J.Q., S.U.A. edited the manuscript with input from all authors.

## Competing interests

The authors declare no competing interests.

## Additional information

**Supplementary information** The online version contains Supplementary material available at https://doi.org/10.1038/s41467-026-70504-1.

