## [Transparent Peer Review File · Nature Communications]

RHD6LA regulates root hair responses to both symbionts and commensals

Corresponding Author: Dr Francesca Tedeschi

Version 0:

Reviewer comments:

Reviewer #2

(Remarks to the Author)

The manuscript, "RHD6LA regulates root hair responses to both symbionts and commensals" by Francesca Tedeschi and colleagues describes a single-cell dataset derived from *Lotus japonicus* plants that were infected by a synthetic community of commensal bacteria (SynCom19), SynCom19 + a nodule-inducing rhizobium, R7A, or mock-treated. The authors cluster and annotate their single-cell data and identify a subcluster of root hair cells that are responsive to the synthetic community (largely defined by the expression of a gene, RHD6LA, homologous to the RSL4 gene from *Arabidopsis*). They purport to show that these cells behave similarly to a cluster of root hair cells from cyclops mutant plants responding to R7A. Lastly, they describe a root hair and nodulation phenotype of *rhhd6la* mutants in response to the synthetic community or R7A infection, respectively. Overall, I found the premise of this study intriguing and it would make an important contribution to the plant / microbial interaction field. It describes novel genetic elements that contribute to commensal colonization. The single-cell dataset will prove a useful resource for others studying this (and potentially other) problems. However, I found the experimental design, analysis, and presentation somewhat lacking in this work. Specific comments are:

1. The authors describe a set of cyclops single-cell datasets, however these are not described in the methods or supplemental data (supplementary data, "cell_ranger_seq"). Are these from a different study?
2. The authors did not include a table with cell barcodes associated with author-defined cell type or other annotations. This is critically important if this dataset will be used as a resource. Furthermore, I could not find the ENA project, "PRJEB89066" on ENA or elsewhere, so cannot assess whether this table or other metadata is available in a publicly accessible resource.
3. The authors hint during the introduction that their synthetic community, SynCom19 promotes plant growth (by stating that community members carry genes with PGP functions), but do not assess whether this community promotes plant growth. I think this is important to assess if they are introducing this as a new resource. Authors should also assess the relative abundance of community members in the plant once colonized – do all 19 members colonize? Are there some members that are dominant? Is 5dpi a relevant time point, or are there microbial dynamics still going on? This may be tangential to their main research question but I think it's an important point to address.
4. From the supplementary table, I gather that there are 2 biological replicates performed, however the authors did not assess how reproducible their transcriptome data are.
5. Line 101: "NPL" is not defined.
6. Line 104: authors describe a "well-defined" cluster containing infected cortical cells, represented in mock- and SynCom19 samples. What constitutes "well-defined"? I'm confused about how this cluster can represent infected cortical cells yet also contain mock-infected cells. How many cells of each population is contained in this cluster?
7. In my opinion, the authors rely too heavily on UMAP representations of their data throughout their manuscript to illustrate their points, where clustering and dot / violin / box plots might be more informative. Their UMAP plots may obscure results due to overplotting, difficulty in distinguishing differences in color scales, and having to examine side-by-side plots to see their points. Authors should reconsider their choice of plots for representing their main points to improve clarity.

8. Authors also present several different combinations of integrated objects in their manuscript (I think 3). For simplicity, the authors might consider creating a single integrated object with all the data under consideration, and parse this larger object into smaller subsets for more focused questions.

9. Line 116-117: "We identified a set of 23 genes..." what clusters / cell types / comparisons were you considering for this? As this gene set is a pivotal one that calls out RHD6LA, I think it's important to clearly define how they got here.

10. Line 123: "...both treatments induce RHD6LA in root hairs" I think this is difficult to ascertain from Fig. 2b. First, "Root Hair" cells in the integrated object presented is not described (no UMAP with cell type annotations), so it's impossible to understand where the subpopulation of cells called out in F2b sits within the larger population of root hair cells. Second, it's hard to say whether the data in F2b is overplotted, and thus how many cells in each population are expressing RHD6LA. Third, how can the authors define an analogous set of cells in the Mock community to compare to? How do they know (in this subpopulation) that this gene is induced? From Fig3a-b it seems that this population of infected root hair cells is a small subpopulation of root hairs in general – what does a mock-treated cell that is within this population mean?

11. The correlation analysis trying to relate R7A-infected vs. cyclops vs. SynCom19 seems very contrived, and difficult to follow, with not much information learned. The overall question the authors are hoping to address is not clearly articulated. The clusters used to set up this analysis are not sufficiently described (there is no figure calling out these clusters, and only a passing description in how they are defined – e.g. "cells showing an aberrant response to R7A").

12. The description of root hair phenotypes was interesting. However Fig. 4d does not include important controls (uninfected WT and mutant roots, R7A-infected WT and mutant roots).

13. Analysis code was not included.

Reviewer #3

(Remarks to the Author)

This study showed that root hairs exhibit overlapping transcriptional responses to symbionts and commensals by single-cell RNA sequencing. Further analysis found that RHD6LA, a transcription factor, regulates root hair responses to both symbiotic and commensal bacteria, including facilitating infection thread formation in response to rhizobia and preventing exaggerated root hair responses to commensal soil bacteria. However, the key scientific question and logic is not clear in the introduction part. Why the authors designed SynCom with and without rhizobia is not clear (line 61: SyCom with rhizobia still has commensal bacterial too). The significance of the major findings is not evident.

1. In Fig 1d, why NPL marker gene is expressed in very different cell clusters? If the authors believe this gene is important, they should delve into these cells for co-expressed genes, which usually helps identify really novel regulators. In line 106, it is not clearly explained that why the data suggest "there was no R7A contamination"?

2. What is the key scientific question, what is the gap in the field? Their key finding and research rationale is not clearly stated in the introduction. Moreover, the conclusion that root hair development regulators affect root microbe interaction is not very novel and surprising.

3. Does RHD6LA directly affect nodule formation in Fig 4, or does it affect root hair development first, and then indirectly affect nodule numbers?

4. Line 126 and fig 3b, I cannot see "consistently" RHD6LA expression in mock group?

5. In "116 We identified a set of 23 genes (Source Data "Common_genes") that were induced by both SynCom19 and R7A., How about the function of other genes? Also, for two scRNAseq dataset (usually 1-10 K DEGs?), it seems impossible only has 23 shared DEGs? How did the authors compare overlapped genes? In which cell type? The manuscript is a bit hard to understand.

6. Line 131, are there grammar issues here? Not easy to understand.

7. The authors should consider using GO enrichment analysis for a lot of places to better illustrate gene functions, for instance, 325 marker genes for SynCom19_RHsubcluster7, 422 for Infected_RH_5dpi or 149 for RH_cyclops_5 cells. That will provide more and critical functional insights for those pathways.

8. Line "103 (Fig. 2a,d)" should be (Fig. 1a,d)

9. Line 172, what is "LORE1"?

Version 1:

Reviewer comments:

Reviewer #2

(Remarks to the Author)

Authors have addressed my concerns

Reviewer #3

(Remarks to the Author)

Thanks for the revision.

* In the final model, The authors concluded that “suggests that CYCLOPS could play a role in suppressing immune signaling”. Essential evidence about the expression of immune genes must be provided for this model. Fig 5 f-h are not supported by data analyses.

* “To identify genes induced by both R7A alone and SynCom19 without R7A; can the authors show the Venn diagram of cell type specific DEGs? The authors concluded that there are “Overlapping root hair transcriptional responses to symbionts and commensals”. This could be misleading, if only very small amount of total DEGs are shared by symbionts and commensals.

*Line 32, the scientific question of “sculpt and establish bacterial communities” seems to be related to microbiome composition, which does not accurately match this manuscript.

* “This response is similar to the rhizobium response in the CSSP-deficient cyclops mutant”, are there any explanation about why it is more similar in mutant relative to WT plants?

**“The swollen and other abnormal root hair phenotypes observed in the presence of commensals were significantly more frequent in the *rhd6la* mutants”; The authors should test each (or most) of the 19 SynCom members, to confirm whether *rhd6la* is really broadly “controls root hair response to commensals and symbionts”.

* Are there huge differences of root responses to pure R7A and SynCom19+R7A? Are there any overall discussion about whether previous mono-association studies in the field has any limitations? Or whether they are basically very similar? Maybe cell type specific DEGs-Venn diagram can be used to reflect that.

**“This is the first time plant responses to a SynCom of commensals has been described at the single cell level.” Are there any biological insights gained from this SynCom -sc-RNAseq study? SynComs could be cool for some context, but could also mess up accurate mono-association conclusions.

Version 2:

Reviewer comments:

Reviewer #3

(Remarks to the Author)

The authors addressed my previous comments.

We wish to thank the reviewers for their thorough evaluation of our manuscript and for their detailed and constructive comments, which have been a great help in guiding the revisions.

REVIEWER COMMENTS

Reviewer #2 (Remarks to the Author):

The manuscript, “RHD6LA regulates root hair responses to both symbionts and commensals” by Francesca Tadeschi and colleagues describes a single-cell dataset derived from *Lotus japonicus* plants that were infected by a synthetic community of commensal bacteria (SynCom19), SynCom19 + a nodule-inducing rhizobium, R7A, or mock-treated. The authors cluster and annotate their single-cell data and identify a subcluster of root hair cells that are responsive to the synthetic community (largely defined by the expression of a gene, RHD6LA, homologous to the RSL4 gene from *Arabidopsis*). They purport to show that these cells behave similarly to a cluster of root hair cells from cyclops mutant plants responding to R7A. Lastly, they describe a root hair and nodulation phenotype of *rhd6la* mutants in response to the synthetic community or R7A infection, respectively. Overall, I found the premise of this study intriguing and it would make an important contribution to the plant / microbial interaction field. It describes novel genetic elements that contribute to commensal colonization. The single-cell dataset will prove a useful resource for others studying this (and potentially other) problems. However, I found the experimental design, analysis, and presentation somewhat lacking in this work. Specific comments are:

1. The authors describe a set of cyclops single-cell datasets, however these are not described in the methods or supplemental data (supplementary data, "cell_ranger_seq"). Are these from a different study?

Response to 1

Yes, these are indeed from a previous study, which we have referenced in the manuscript. We have now stated this more clearly in the text to avoid confusion.

2. The authors did not include a table with cell barcodes associated with author-defined cell type or other annotations. This is critically important if this dataset will be used as a resource. Furthermore, I could not find the ENA project, “PRJEB89066” on ENA or elsewhere, so cannot assess whether this table or other metadata is available in a publicly accessible resource.

Response to 2

That is a very good point. We have included the metadata table in the Source Data for easy access and made the ENA project “PRJEB89066” publicly available. In addition, we shared the RDS files on FigShare for easy re-use.

3. The authors hint during the introduction that their synthetic community, SynCom19 promotes plant growth (by stating that community members carry genes with PGP functions), but do not assess whether this community promotes plant growth. I think this is important to

assess if they are introducing this as a new resource. Authors should also assess the relative abundance of community members in the plant once colonized – do all 19 members colonize? Are there some members that are dominant? Is 5dpi a relevant time point, or are there microbial dynamics still going on? This may be tangential to their main research question but I think it's an important point to address.

Response to 3

We agree that while it is not critical for the main conclusions, this information would be useful to promote further use of SynCom19. We have now added data on the microbiome composition as requested (main text, lines 78-84) and Supplementary figure 2), indicating that the plant rhizosphere is well-colonized by all members of SynCom19 at 5 dpi. We have also assessed the impact of SynCom19 inoculation on plant growth in another independent experiment in plate, using the same setting as for the single cell RNA-seq experiment. We found that SynCom19 alone did not affect plant biomass, whereas inoculation with R7A increased biomass relative to mock and SynCom19 treatments. Co-inoculation with SynCom19 and R7A resulted in a similar or slightly enhanced growth promotion compared to R7A alone. We have added these results as a new Supplementary figure (Supplementary figure 3).

4. From the supplementary table, I gather that there are 2 biological replicates performed, however the authors did not assess how reproducible their transcriptome data are.

Response to 4

Indeed, we used two biological replicates and have now emphasized this in the text. To visualize reproducibility, we have added supplementary figures with UMAPs colored by replicate (Supplementary figure 4)

5. Line 101: “NPL” is not defined.

Response to 5

We have defined *NPL* as *Nodule Pectate Lyase*.

6. Line 104: authors describe a “well-defined” cluster containing infected cortical cells, represented in mock- and SynCom19 samples. What constitutes “well-defined”? I’m confused about how this cluster can represent infected cortical cells yet also contain mock-infected cells. How many cells of each population is contained in this cluster?

Response to 6

We agree that this could have been stated more clearly. Well-defined, in this case, simply refers to a cluster detected by the clustering algorithm. We have now written this instead of well-defined. The cluster contains 38 cells from the mock inoculated, 187 cells from SynCom19 inoculated and 134 cells from SynCom19+R7A inoculated samples. Only the cells from SynCom19+R7A inoculated samples show *NPL* expression in that cluster. We have included this information in the text. We have also added a list of marker genes for the

SynCom19+R7A cluster in question as well as a dot plot to illustrate the expression of nodulation genes only in the SynCom19+R7A samples (Figure 1d).

7. In my opinion, the authors rely too heavily on UMAP representations of their data throughout their manuscript to illustrate their points, where clustering and dot / violin / box plots might be more informative. Their UMAP plots may obscure results due to overplotting, difficulty in distinguishing differences in color scales, and having to examine side-by-side plots to see their points. Authors should reconsider their choice of plots for representing their main points to improve clarity.

Response to 7

We have favored UMAPs here because it was otherwise difficult to illustrate the responses in specific root hair populations. To address possible issues with overplotting and facilitate comparison across maps, we have added a new panel d) to figure 3, where we zoom in on the relevant populations and compare them directly with *RHD6LA* expression. We have also used a dot plot in Figure 1.

8. Authors also present several different combinations of integrated objects in their manuscript (I think 3). For simplicity, the authors might consider creating a single integrated object with all the data under consideration, and parse this larger object into smaller subsets for more focused questions.

Response to 8

It is true that it would be attractive with a single integrated object. However, for the sake of the narrative, we think it is important to start out with an object that only contains data from the current study before expanding with additional data from published work. We only have two different integrated objects.

9. Line 116-117: “We identified a set of 23 genes...” what clusters / cell types / comparisons were you considering for this? As this gene set is a pivotal one that calls out *RHD6LA*, I think it’s important to clearly define how they got here.

Response to 9

We agree that this should be more carefully explained. It is now clearly stated in the main manuscript text: “We identified 23 genes that were induced both by SynCom19 5dpi in our scRNA-seq dataset and by R7A at 10dpi in the published Lotus wild-type scRNA-seq dataset³⁰. Genes were defined as common based on differential expression across all cell types ($p_{\text{adj}} \leq 0.05$ and $\text{pct.2} \leq 0.02$) (Source Data “Common_genes”).”

10. Line 123: “...both treatments induce *RHD6LA* in root hairs” I think this is difficult to ascertain from Fig. 2b. First, “Root Hair” cells in the integrated object presented is not described (no UMAP with cell type annotations), so it’s impossible to understand where the subpopulation of cells called out in F2b sits within the larger population of root hair cells.

Second, it's hard to say whether the data in F2b is overplotted, and thus how many cells in each population are expressing RHD6LA. Third, how can the authors define an analogous set of cells in the Mock community to compare to? How do they know (in this subpopulation) that this gene is induced? From Fig3a-b it seems that this population of infected root hair cells is a small subpopulation of root hairs in general – what does a mock-treated cell that is within this population mean?

Response to 10

This is an excellent point and indeed an oversight on our side. We have now added a UMAP annotated with cell types to figure 3. With respect to the population for comparison in the mock inoculated samples, those are the root hair cells in the same cluster. Although they do not show RHD6LA expression, they still share sufficient transcriptional similarity to be in close proximity to the inoculated root hair cells in the UMAP. We have clarified this in the text. In addition, we have added a new panel to Figure 3, zooming in on the relevant root hair populations to alleviate the overplotting issue.

11. The correlation analysis trying to relate R7A-infected vs. *cyclops* vs. SynCom19 seems very contrived, and difficult to follow, with not much information learned. The overall question the authors are hoping to address is not clearly articulated. The clusters used to set up this analysis are not sufficiently described (there is no figure calling out these clusters, and only a passing description in how they are defined – e.g. “cells showing an aberrant response to R7A”).

Response to 11

We are sorry to hear that this comes across as contrived. This link actually emerged from simply browsing expression patterns in UMAPs across studies, where we noticed a very clear similarity between the SynCom19 responses in the wild type and the R7A responses in the *cyclops* mutant, most often with exaggerated expression in *cyclops*. The correlation analysis was an attempt at quantifying these effects and indeed showed statistical significance. To illustrate the point graphically as well, we have restructured figure 3 so Fig. 3d now shows the three root hair populations in the integrated object. The reason that we did not provide further details on the *cyclops* root hair population with aberrant responses, is that this was described in Frank et al., 2023. We have now added a sentence to clarify this and provide a brief description and the cell number of each population at first mention. We have also refined the set of marker genes for the three populations, requiring support from a larger number of cells (see reviewer 3 point 7). To add further support to the conclusions, we have carried out a simpler correlation analysis based on the expression of all genes (Supplementary figure 5). Regardless of using either of the marker gene sets or all genes in the correlation analysis, the conclusion remained the same. We have underlined the importance of the correlation results in the main text by stating that “the correlation between RH_ *cyclops*_5 and SynCom19_RHsubcluster7 cells was higher than that between RH_ *cyclops*_5 and Infected_RH_5dpi (P = 0.0002, Pearson and Filon's z (1898) test), although both these populations were inoculated with R7A and were assayed in the same experiment”.

12. The description of root hair phenotypes was interesting. However Fig. 4d does not

include important controls (uninfected WT and mutant roots, R7A-infected WT and mutant roots).

Response to 12

We had not included those to keep the figure simple, only showing the different root hair phenotypes and presenting the quantitative data in the other panels. We are happy to add additional images and have now done so.

13. Analysis code was not included.

Response to 13

We have added a link to the analysis code on GitHub.

Reviewer #3 (Remarks to the Author):

This study showed that root hairs exhibit overlapping transcriptional responses to symbionts and commensals by single-cell RNA sequencing. Further analysis found that RHD6LA, a transcription factor, regulates root hair responses to both symbiotic and commensal bacteria, including facilitating infection thread formation in response to rhizobia and preventing exaggerated root hair responses to commensal soil bacteria. However, the key scientific question and logic is not clear in the introduction part. Why the authors designed SynCom with and without rhizobia is not clear (line 61: SyCom with rhizobia still has commensal bacterial too). The significance of the major findings is not evident.

Response

Thank you for presenting this critical perspective on our work. From your comments, it is clear that we generally need to more carefully clarify the research questions and significance of the findings. We have now implemented these changes in the manuscript.

With respect to designing the SynCom with and without rhizobia. Please note that SynCom19 does not contain a symbiotic rhizobium. This design was chosen because rhizobial infection of plants generally takes place in the presence of other microbes, so our experimental setup comes closer to the natural situation, and because we had already generated data in a previous study for inoculation with R7A alone. We have clarified this in the main text (lines 76-78).

Please find the detailed point-by-point response to all queries below.

1. In Fig 1d, why NPL marker gene is expressed in very different cell clusters? If the authors believe this gene is important, they should delve into these cells for co-expressed genes, which usually helps identify really novel regulators. In line 106, it is not clearly explained why the data suggest “there was no R7A contamination”?

Response to 1

Thank you for pointing this out. We realise now that our origins in the symbiotic nitrogen fixation field led us to omit some important details here. We have now explained that *NPL*, *NODULATION PECTATE LYASE*, is a marker gene in both root hair and cortical cells, which is why it shows up as expressed in multiple clusters. We have also explained that *NPL* is highly specific to rhizobium infection and that this is why its absence in the SynCom19 inoculated samples indicates that there was no contamination with bona fide, nod factor producing symbionts. Our purpose is not to study already well-described nodulation genes, so we will not investigate *NPL* further here. We have also added a more comprehensive analysis of known symbiosis genes in figure 1d to strengthen the point that there was no contamination with R7A symbiotic rhizobia and thus no induction of symbiotic signalling.

2. What is the key scientific question, what is the gap in the field? The key finding and research rationale is not clearly stated in the introduction. Moreover, the conclusion that root hair development regulators affect root microbe interaction is not very novel and surprising.

Response to 2

This is the first time plant responses to a SynCom of commensals has been described at the single cell level. Therefore, the initial research question was pretty open - simply how plant roots respond to commensals at the single cell level and what the overlap might be with the already relatively well-studied response to specialised symbionts. We have edited the introduction to clarify. With respect to the significance, we do not claim that *RHD6LA* is a root hair developmental regulator. Indeed, the aberrant *rhid6la* mutant phenotype is contingent on microbial inoculation, whereas we see no root hair developmental phenotypes in the absence of bacteria. We realise now that this was not explicitly stated, and have now highlighted this in the *RHD6LA* results section (lines 200-202). We believe it is clear from the abstract and discussion that the overlap between commensal and symbiotic signaling in a specific subset of root hairs and the role of *RHD6LA* in regulating both interactions represent significant and novel findings. We are happy to add further clarification if needed, and welcome specific suggestions.

3. Does *RHD6LA* directly affect nodule formation in Fig 4, or does it affect root hair development first, and then indirectly affect nodule numbers?

Response to 3

Thank you for raising this point. Because *RHD6LA* expression is confined to root hairs and absent from nodules or primordia, our findings indicate that the lower nodule numbers arise indirectly, as a consequence of impaired root hair development and reduced infection thread formation, rather than from a direct role of *RHD6LA* in nodule organogenesis. We have added this consideration to the results (lines 211-214).

4. line 126 and fig 3b, I cannot see “consistently” *RHD6LA* expression in mock group?

Response to 4

This appears to be a misunderstanding. We have not claimed that RHD6LA has a consistently expression in mock but just that “only subcluster 7, comprising 92 out of 1658 root hair cells, consistently expressed RHD6LA in the SynCom19-treated samples (lines 148-149)

5. In ”116 We identified a set of 23 genes (Source Data “Common_genes”) that were induced by both SynCom19 and R7A., How about the function of other genes? Also, for two scRNAseq dataset (usually 1-10 K DEGs?), it seems impossible only has 23 shared DEGs? How did the authors compare overlapped genes? In which cell type? The manuscript is a bit hard to understand.

Response to 5

We have added additional text to make this part easier to follow: “We identified 23 genes that were induced both by SynCom19 5dpi in our scRNA-seq dataset and by R7A at 10dpi in the published Lotus wild-type scRNA-seq dataset³⁰. Genes were defined as common based on differential expression across all cell types ($p_{\text{adj}} \leq 0.05$ and $\text{pct.2} \leq 0.02$) (Source Data “Common_genes”).”

6. Line 131, are there grammar issues here? Not easy to understand.

Response to 6

We revised for clarity: “Wild type SynCom19 responses resemble those of *cyclops* to R7A”

7. The authors should consider using GO enrichment analysis for a lot places to better illustrate gene functions, for instance, 325 marker genes for SynCom19_RHsubcluster7, 422 for Infected_RH_5dpi or 149 for RH_cyclops_5 cells. That will provide more and critical functional insights for those pathways.

Response to 7

Yes, this is a good idea. We performed GO enrichment analyses as suggested on the genes identified in the three populations (Infected_RH_5dpi, SynCom19_RHsubcluster7, and RH_cyclops_5). This analysis revealed that several of the genes under consideration were expressed in only one cell in the Infected_RH_5dpi population. Based on this observation, we refined our correlation analysis (Fig. 3e–f) by adjusting the expression filter to $\text{pct.1} \geq 0.1$ and $\text{pct.2} \leq 0.1$ to ensure a more robust analysis. We then repeated the GO enrichment analysis using the updated set of marker genes. However, this resulted mainly in only one or two significant genes per category (see tables below), leaving us uncertain about the reliability. Therefore, we decided not to include the GO enrichment results in the manuscript at this stage.

GO enrichment Infected_RH_5dpi

GO.ID	Term	Annotated	Significant	Expected	weightFisher
GO:0006857	oligopeptide transport	18	2	0,1	0,0043
GO:0090378	seed trichome elongation	3	1	0,02	0,0164
GO:0015969	guanosine tetraphosphate metabolic process	4	1	0,02	0,0218
GO:0019441	tryptophan catabolic process to kynurenine	4	1	0,02	0,0218
GO:1901601	strigolactone biosynthetic process	4	1	0,02	0,0218
GO:0010105	negative regulation of ethylene-activated signaling pathway	6	1	0,03	0,0326
GO:0001578	microtubule bundle formation	6	1	0,03	0,0326
GO:0007188	adenylate cyclase-modulating G protein-coupled receptor signaling pathway	7	1	0,04	0,0379
GO:0070525	tRNA threonylcarbamoyladenine metabolic process	9	1	0,05	0,0485

GO enrichment SynCom19_RHsubcluster7

GO.ID	Term	Annotated	Significant	Expected	weightFisher
GO:1901601	strigolactone biosynthetic process	4	2	0,03	0,00025
GO:0120009	intermembrane lipid transfer	8	2	0,05	0,00115
GO:0080167	response to karrikin	10	2	0,07	0,00183
GO:0009439	cyanate metabolic process	1	1	0,01	0,00652
GO:0071918	urea transmembrane transport	1	1	0,01	0,00652
GO:0032447	protein urmylation	2	1	0,01	0,01299
GO:0042744	hydrogen peroxide catabolic process	82	3	0,53	0,01637
GO:0032264	IMP salvage	3	1	0,02	0,01942
GO:1902074	response to salt	3	1	0,02	0,01942
GO:0010116	positive regulation of abscisic acid biosynthetic process	3	1	0,02	0,01942
GO:0009695	jasmonic acid biosynthetic process	5	1	0,03	0,03217
GO:0031146	SCF-dependent proteasomal ubiquitin-dependent protein catabolic process	6	1	0,04	0,03847
GO:0034227	tRNA thio-modification	6	1	0,04	0,03847
GO:0008152	metabolic process	10576	66	68,92	0,03977

GO enrichment RH_cyclops_5

GO.ID	Term	Annotated	Significant	Expected	weightFisher
GO:1901601	strigolactone biosynthetic process	4	2	0,02	0,00019
GO:0006952	defense response	165	7	0,94	0,00051
GO:0055074	calcium ion homeostasis	11	2	0,06	0,00565
GO:0009788	negative regulation of abscisic acid-activated signaling pathway	2	1	0,01	0,01141
GO:1902289	negative regulation of defense response to oomycetes	2	1	0,01	0,01141
GO:2000071	regulation of defense response by callose deposition	2	1	0,01	0,01141
GO:0010116	positive regulation of abscisic acid biosynthetic process	3	1	0,02	0,01706
GO:0002238	response to molecule of fungal origin	3	1	0,02	0,01706
GO:1902074	response to salt	3	1	0,02	0,01706
GO:0009607	response to biotic stimulus	161	4	0,92	0,02023
GO:0051209	release of sequestered calcium ion into cytosol	4	1	0,02	0,02269
GO:0019441	tryptophan catabolic process to kynurenine	4	1	0,02	0,02269
GO:1900150	regulation of defense response to fungus	5	1	0,03	0,02828
GO:0009695	jasmonic acid biosynthetic process	5	1	0,03	0,02828
GO:0009738	abscisic acid-activated signaling pathway	16	2	0,09	0,03344
GO:0040013	negative regulation of locomotion	7	1	0,04	0,03937
GO:0010200	response to chitin	8	1	0,05	0,04487

8.Line “103 (Fig. 2a,d)” should be (Fig. 1a,d)

Response to 8

Thank you for pointing this out. We have revised the figure references.

9.Line 172, what is “LORE1” ?

Response to 9

A *LORE1* line is a *Lotus japonicus* plant line in which the endogenous *LORE1* retrotransposon has been inserted into a specific gene, creating a stable insertional mutant. In addition to the reference: Małolepszy, A. *et al.* The *LORE 1* insertion mutant resource. *The Plant Journal* **88**, 306–317 (2016), we have now also briefly mentioned this in the main text (in the Methods lines 307-309).

Thank you for the additional comments. When carefully revising the manuscript, we noticed a mistake in the initial gene list (Source Data “Common_genes”), which contained six genes that were selected in a way not described in the manuscript. We have now corrected the list and clarified how the initial candidates were selected, including the relevant Venn diagram in the supplementary figure 5. Specifically, since our aim was to assess whether transcriptional responses to symbiotic and commensal bacteria overlap in cell types associated with infection processes, such as root hair and nodule cells, we examined the overlap between genes upregulated by the SynCom19 at 5 dpi and the *M. loti* R7A-associated markers. Specifically, we compared SynCom19-induced genes with R7A infected cell markers and R7A nodule cell markers identified at 10 dpi (Frank, Fechete et al., 2023), using thresholds of adjusted $p \leq 0.05$ and $\text{pct.2} \leq 0.02$. From that comparison we were able to find 17 common genes. (Source Data “Infected_markers_10dpi”, Nodule_markers_10dpi and “Common_genes”) (manuscript lines 130-143).

Thanks for the revision.

* In the final model, The authors concluded that “suggests that CYCLOPS could play a role in suppressing immune signaling”. Essential evidence about the expression of immune genes must be provided for this model. Fig 5 f-h are not supported by data analyses.

Response to 1

We agree that we do not provide evidence to conclusively select one of the scenarios presented in figure 5. This figure represents alternative scenarios to help inspire discussion and further research in the field. We have indicated this by entitling the figure “Scenarios for root hair responses to commensal and symbiotic bacteria.” We have also clarified the limitations by stating that “The scenarios presented in Fig. 5 are simplifications. In reality, Nod factors, conserved microbe-associated molecular patterns, microbial effectors and plant detection systems result in a great variation in interaction outcomes, especially when no fully compatible symbionts are available....”. To further reduce the risk of misunderstandings, we have now stated that “Here, we explore different scenarios for how the observed transcriptional overlap could be generated.” at an earlier point in the discussion and explicitly mentioned the word “scenario” also in the main text when relevant.

* “To identify genes induced by both R7A alone and SynCom19 without R7A; can the authors show the Venn diagram of cell type specific DEGs? The authors concluded that there are “Overlapping root hair transcriptional responses to symbionts and commensals”. This could be misleading, if only very small amount of total DEGs are shared by symbionts and commensals.

Response to 2

To clarify the selection process, we have now added two Venn diagrams in Supplementary figure 5. The first shows a large overlap in the overall transcriptional responses to SynCom19 and R7A when viewed globally across all clusters. The second shows the more specific comparison to genes identified as specific markers for rhizobium-responsive cells. We have also clarified this in the manuscript text. This initial analysis revealed two shared

root hair-specific genes between R7A alone and SynCom19 in the absence of R7A. We followed up with a more detailed analysis (manuscript lines 167-171) of the relevant subset of root hair cells, including an UpSet plot (Fig. 3e), which provides an alternative representation of a Venn diagram for multiple gene sets. This analysis revealed a substantial overlap of root hair differentially expressed genes between R7A alone (infected root hairs at 5 dpi) and SynCom19-treated samples (SynCom19 RHsubcl 7), comprising 21 shared genes. An even larger overlap was observed between the R7A *cyclops* condition and SynCom19, with 47 shared genes. These results support the conclusion that symbiotic and commensal bacteria elicit overlapping transcriptional responses in root hair cells, while also clarifying that such overlap becomes evident only upon cell type-specific analysis.

*Line 32, the scientific question of “sculpt and establish bacterial communities” seems to be related to microbiome composition, which does not accurately match this manuscript.

Response to 3

We thank the reviewer for this comment. The statement in question appears in the Introduction and was not intended to imply that we directly addressed this aspect in the present study. However, we agree that the original wording could be interpreted as suggesting an analysis of microbiome composition, which is not the primary focus of this manuscript. We have therefore revised the text to more accurately reflect the scope of our work. The revised sentence now reads: “However, despite the importance of the root microbiota, the molecular mechanisms by which plants perceive and respond to commensal bacteria at the level of individual root cell types remain poorly understood”.

* “This response is similar to the rhizobium response in the CSSP-deficient *cyclops* mutant”, are there any explanation about why it is more similar in mutant relative to WT plants?

Response to 4

The likely explanation is that both SynCom19-treated WT root hairs and R7A-treated *cyclops* root hairs experience bacterial perception without progression into infection thread formation. WT + R7A root hairs, in contrast, activate the full symbiotic program downstream of CSSP/CYCLOPS, diverging from the SynCom19 response. We have added a paragraph discussing this logic (manuscript lines 250-256).

**The swollen and other abnormal root hair phenotypes observed in the presence of commensals were significantly more frequent in the *rhod6la* mutants”; The authors should test each (or most) of the 19 SynCom members, to confirm whether *rhod6la* is really broadly “controls root hair response to commensals and symbionts”.

Response to 5

We agree that a single strain could be responsible for the *rhod6la* exaggerated hair response. To examine the generality, we have now conducted additional experiments with selected members of the SynCom19 community as suggested. We performed mono-inoculation experiments with four strains representing four different bacterial families (Comamonadaceae, Flavobacteriaceae, Pseudomonadaceae, and Burkholderiaceae). These experiments showed that three of the four tested strains (LjR62, LjR82 and LjR296),

induced a higher frequency of aberrant root hair phenotypes in the *rhd6la* mutant compared with the wild type, proving that this effect is not limited to a single strain, but represents a more general response to commensals. (manuscript lines 223-230).

* Are there huge differences of root responses to pure R7A and SynCom19+R7A? Are there any overall discussion about whether previous mono-association studies in the field has any limitations? Or whether they are basically very similar? Maybe cell type specific DEGs-Venn diagram can be used to reflect that.

Response to 6

A detailed comparison of R7A alone and SynCom19+R7A would also be interesting, but is not the main theme of this manuscript, since we are focusing on the novel aspects of responses to commensals and their similarities to the symbiotic responses, not how the presence of commensals affects symbiotic signaling.

Mono-inoculation and SynCom inoculation each have their advantages and disadvantages. Mono-inoculation directly links a single strain to a specific response but cannot capture combinatorial community interactions impacting the plant response. The reverse is true for SynCom inoculation. Both are, in our opinion, relevant approaches for understanding plant-microbe interactions. We have now cited a recent Arabidopsis paper describing responses to microbes at the single-cell level and briefly mentioned these general considerations in the discussion. (manuscript line 59 and 236-240).

*“This is the first time plant responses to a SynCom of commensals has been described at the single cell level.” Are there any biological insights gained from this SynCom -sc-RNAseq study? SynComs could be cool for some context, but could also mess up accurate mono-association conclusions.

Response to 7

Plants never interact with single pure strains in their natural environments, so this study is at least one step closer to the natural situation. We have used a community of commensals that do not produce nod factors but are very different in many other respects and capture many different bacterial functionalities. That allows us clearly to distinguish symbiotic from non-symbiotic signaling, which is the main biological point of the paper, leading to the novel insight that there are specific bacterial perceptions systems active in a limited population of root hairs - insight only possible with single-cell resolution analysis. It is true that we did not disentangle the effect of the community as whole from the actions of a potentially dominant single strain responsible for all the effects seen. We have now addressed this issue using the mono-inoculation and root hair deformation results, which shows that multiple strains induce root hair morphological changes in a *rhd6la* dependent manner. We have now added the data to the manuscript and clarified their significance in the results and discussion sections.